# Design of a 2 m Primary Mirror Assembly Considering Fatigue Characteristics

Jiakun Xu [1,2], Wei Li [1,2,*], Kejun Wang [1], Qu Hao [1] and Guanyu Lin [1]

1 Changchun Institute of Optics, Fine Mechanics and Physics, Chinese Academy of Sciences, Changchun 130033, China
2 Center of Materials Science and Optoelectronics Engineering, University of Chinese Academy of Sciences, Beijing 100049, China
* Correspondence: leew2006@ciomp.ac.cn.com

**Abstract:** The design requirements of a 2 m mirror assembly installed on a large space optical remote sensor are investigated in this study. The mirror body and rigid connectors are designed using topology and size optimization methods. The initial design scheme of flexible supports is then proposed through an in-depth exploration of fatigue failure mechanism caused by impacts of size parameters, thermal–mechanical vibration, and surface abrasion. The comprehensive analysis mode of the flexible support structure is established with the help of finite element and fatigue analysis software programs. The designed mirror surface error under composited force is 5.6 nm, first natural vibration mode is 119.16 Hz, and fatigue life synthesizing thermal–mechanical vibration is about 21,790,000 cycles, thereby meeting the design requirements. The first natural vibration mode, acceleration response, and stress responses of sine and random vibration are verified with experiments, and the findings are compared with the theoretical analysis results. The analysis mode can successfully and significantly improve the reliability of the mirror assembly and help optimize the design of flexible supports.

**Keywords:** flexible supports; fatigue life; analysis mode; finite element analysis; fatigue analysis

## 1. Introduction

Large-aperture mirrors are commonly used in ground-based telescopes, and gradually applied to space optical remote sensors with the development of launch vehicle technology. Image quality improves with the increased mirror aperture. The processing difficulty, processing costs, and time increase at the same time. In addition, repairing remote sensors during operation in space is difficult and costs a great amount. Hence, the quality of the mirror assembly must be considered to ensure the service time of remote sensors. Support structures must be flexible during the design stage to eliminate the influences on mirror surface error caused by temperature fluctuation and assembly stress. Grooving and reducing the cross section are effective methods for minimizing the rigidity and releasing designated degrees of freedom. However, a flexible design will partially increase stress and strain while generating stress concentration and causing local plastic deformation on the structure. Stress concentration may cause fatigue failure, reduce the reliability of the optical system, and finally affect the imaging quality of the whole camera [1,2]. The stress concentration $K_T$ and the fatigue notch $K_f$ factors are typically used to describe the effect of stress concentration on fatigue strength under static force or simple alternating load. However, describing the complicated random vibration fatigue life with only these two coefficients is difficult because the flexible structure will bear uncertain random vibration load during experiment, transportation, and launching [3].

A determinate function is unable to describe the excitation and the response of a random vibration, and the instantaneous value is also unpredictable. However, they follow certain statistical laws and can be described using statistical theory. Power spectral density

function (PSD) is usually applied to describe the load of random vibration. Knowing the geometric characteristics of parts, fatigue property of materials, PSD of excitation signal, and surface state of components is necessary during fatigue analysis. Obtaining the time history of stress and strain responses through finite element analysis software, S–N curve of materials, and surface condition of products is the main method for calculating fatigue life [4]. However, neither influences of size parameters nor influences of temperature and residual stress on fatigue life are considered in traditional design methods [5,6]. Fatigue life is analyzed in this work by exploring the failure mechanism. In addition, analysis mode of the flexible structure is established to improve the reliability of the mirror assembly.

## 2. Mirror Assembly Design

### 2.1. Design of Mirror and Rigid Connectors

The 2050 mm major mirror body made of SiC material adopts a passive support form with three points on it back. Three mounting holes with a taper of 1:18 are set at the back of the mirror body. The maximum diameter of the holes is $\phi$172 mm, the depth of the holes is 192 mm, and the distribution diameter of the mounting holes is $\phi$1360 mm. The mirror is shown in Figure 1.

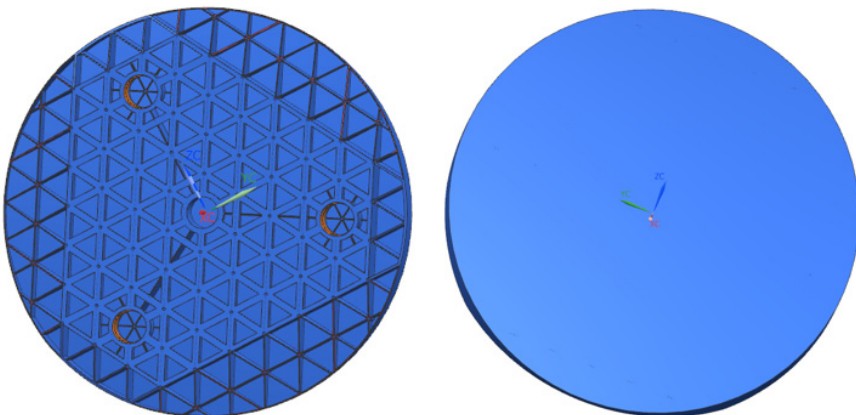

**Figure 1.** Mirror.

Rigid connectors with the same taper of mounting holes are glued together with supporting holes using silicone rubber adhesive. Connectors are made of invar material, which presents the same thermal expansion coefficient as the mirror. The connection method is shown in Figure 2. Six M10 threaded holes are set at the bottom of each rigid connector. The diameter of threaded distribution holes is $\phi$100 Flexible supports are connected to threaded holes using screws, as shown in Figure 3.

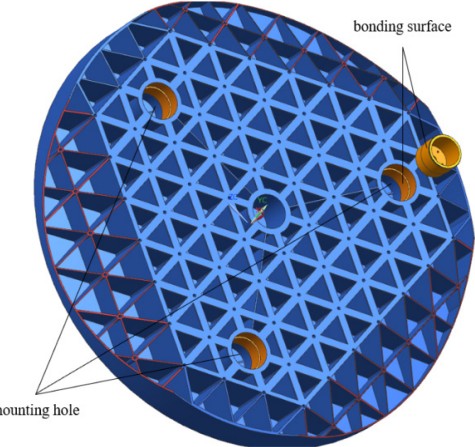

**Figure 2.** Connection method of the mirror body and rigid connector.

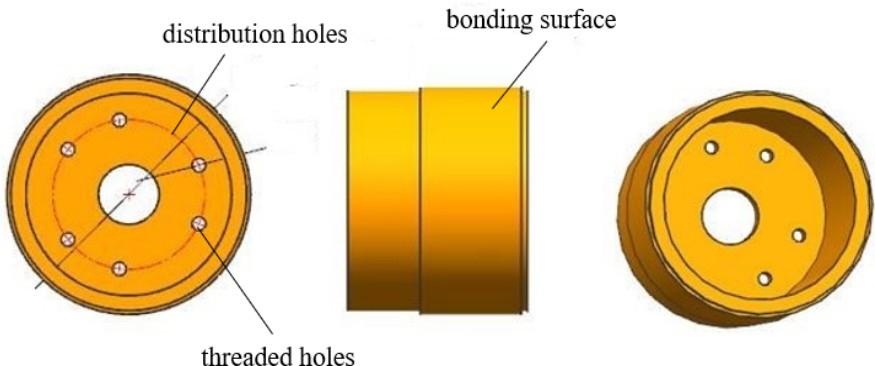

**Figure 3.** Rigid connector.

## 2.2. Design of Flexible Supports

The initial design of the flexible support structure is presented in Figure 4.

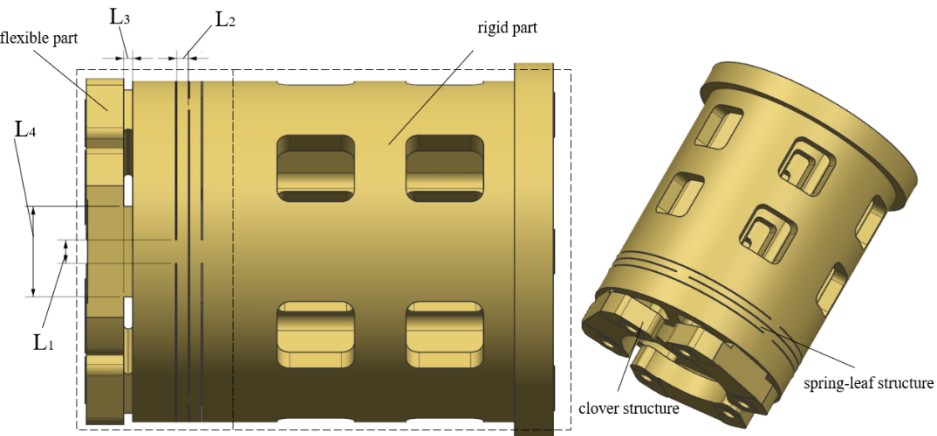

**Figure 4.** Flexible support structure.

The support structure is divided into flexible and rigid parts. The flexible part should be able to move radially and rotate within a certain range to reduce the mirror surface error under gravity, temperature fluctuation, and non-flatness error. At the same time, it should balance the stiffness and flexibility to ensure the fundamental frequency and rigid body displacement of the mirror [7,8]. Therefore, the flexible part is designed as a spring leaf to provide increased rotational flexibility without reducing the axial rigidity. The clover flexible structure connected to the rigid connector can provide axial flexibility by grooving along the central axis. Four key parameters affecting the performance of flexible supports include width of the radial flexible sheet $L_1$, length of the radial flexible sheet $L_2$, length of the axial flexible groove $L_3$, and thickness of the clover support $L_4$. The mirror surface error RMS value and first-order fundamental frequency of mirror assembly are taken as the objective function, and optimal values of the four parameters are determined through size optimization.

## 2.3. Establishment of Flexible Structure Analysis Mode

The analysis mode of the flexible support structure is established to verify the reliability of the mirror assembly. The analysis flow diagram is presented in Figure 5. The mode comprehensively considers the static, dynamic, and fatigue effects on flexible supports.

The mirror body and test tooling are divided by shell elements. The flexible support structures and rigid connectors are mainly divided by octa-node hexahedral elements. The finite element model is shown in Figure 6, and the mesh generation process is presented in Table 1.

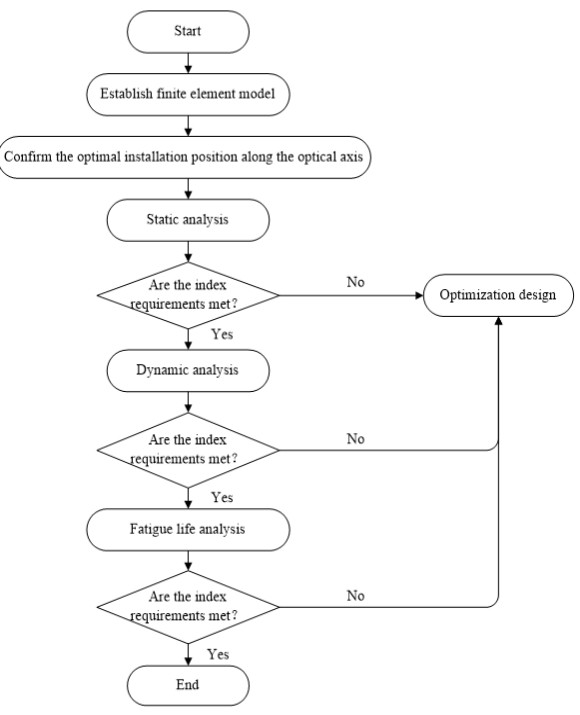

**Figure 5.** Analysis flow diagram.

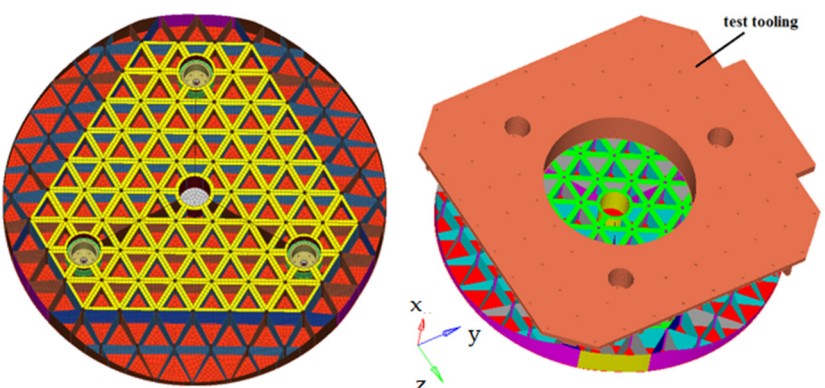

**Figure 6.** Finite element mode.

**Table 1.** Mesh generation of mirror assembly.

| Component | Quantity | Material | Elements | Nodes | Card Image |
|:---:|:---:|:---:|:---:|:---:|:---:|
| **Mirror** | 1 | SiC | 156,476 | 141,724 | PSHELL |
| **Rigid connector** | 3 | 4J32 | 39,932 | 30,810 | PSOLID |
| **Flexible support** | 3 | TC17 | 62,994 | 46,542 | PSOLID |
| **Test tooling** | 1 | Al | 128,341 | 95,566 | PSHELL |
| **Total** | | | 387,743 | 314,642 | |

Test tooling and constraint are removed from the 18 threaded holes on the back of flexible supports during static analysis. The test tooling is connected to the vibration experimental platform, and holes at its bottom are constrained during dynamic analysis. We calculate the mirror surface errors according to the analysis results using MATLAB software. We perform modal, vibration, and fatigue life analyses when the analysis results meet the design requirements. These processes are connected via ISIGHT software, and an automated simulation mode is built to achieve the optimal design of the flexible support. The design route of the mode is shown in Figure 7. According to the finite element analysis

results, the optimal $L_1$, $L_2$, $L_3$, and $L_4$ values are 9, 2.5, 3.5, and 35 mm, respectively. However, the mirror surface error is significantly more than the design indexes. Hence, determining the correct installation position of flexible supports is necessary.

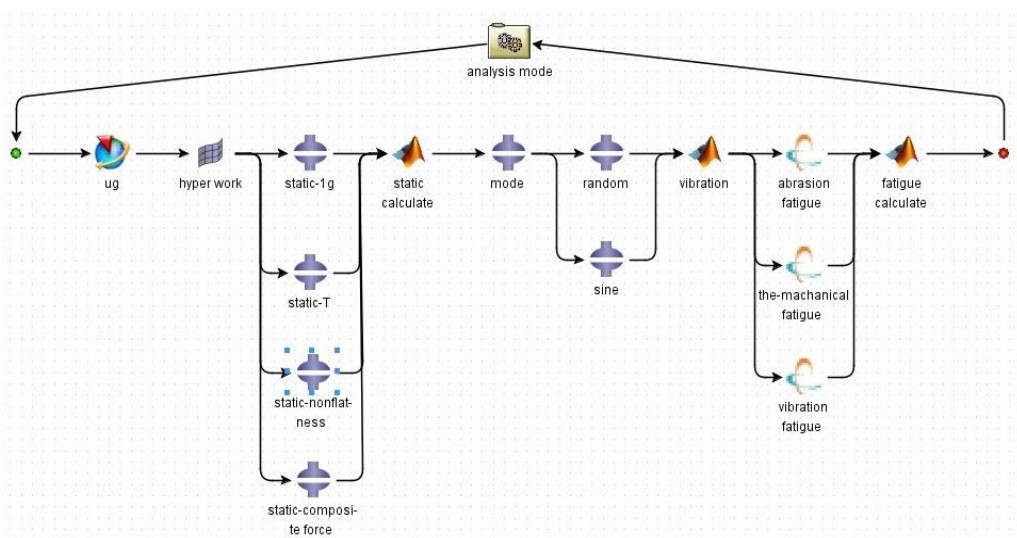

**Figure 7.** Design route in ISIGHT software.

## 3. Installation Position Optimization

The installation position of flexible supports directly affects the imaging of the mirror assembly when the simulation model of the mirror assembly is established. The optimal installation position is determined using neutral plane theory and static analysis.

### 3.1. Neutral Plane Theory

The neutral plane refers to a virtual plane that passes through the barycenter of the mirror and is paralleled to the installation plane of the mirror assembly. As shown in Figure 8, the mirror body is placed vertically, one flexible support is at the top, and two other flexible supports are at the bottom and placed in the same horizontal plane. Each flexible support bears one-third of the total weight of the mirror.

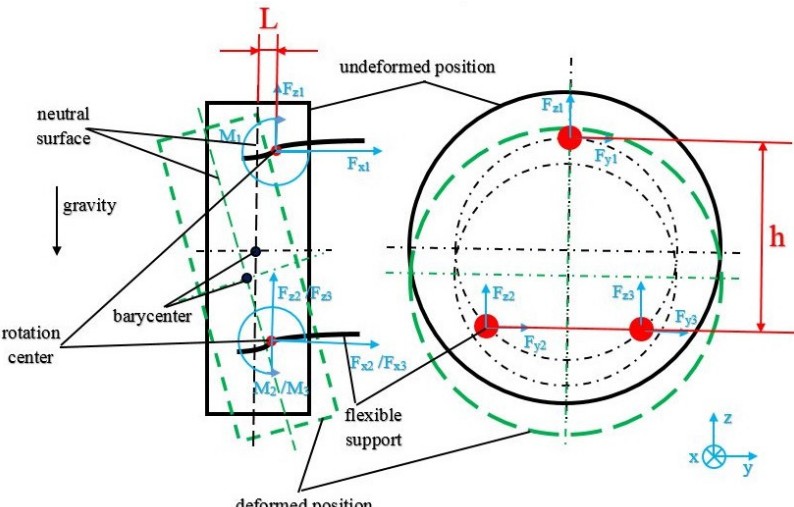

**Figure 8.** Stress condition of the flexible structure when the mirror is placed vertically.

Assuming that mounted surfaces of flexible supports remain unchanged, the mirror acts as the reciprocating motion along the optical axis horizontally by changing the length

of the rigid connector and flexible supports. Moreover, the mirror body rotates around the rotation center of the flexible structure and deforms and tilts in the direction of gravity. Suppose that the distance between the neutral surface and the rotation center is $L$, then stress equations of each flexible structure are expressed as follows:

$$\begin{cases} F_{z1} = F_{z2} = F_{z3} = \dfrac{G}{3} \\ F_{x1} + F_{x2} + F_{x3} = 0 \\ \dfrac{2 \cdot F_{x1} \cdot h}{3} + \dfrac{F_{x2} \cdot h}{3} + \dfrac{F_{x3} \cdot h}{3} + M_1 + M_2 + M_3 = G \cdot L \\ F_{x2} = F_{x3} \\ M_1 = M_2 = M_3 \end{cases} \tag{1}$$

Formula (1) can be simplified as follows:

$$\begin{cases} F_{x1} \cdot h + 3M_1 = G \cdot L \\ F_{x1} = -2F_{x2} = -2F_{x3} \end{cases} \tag{2}$$

The moment caused by gravity will be close to 0 and $M_1$ is theoretically the minimum value when the rotation center is situated on the neutral plane. The axial force $F_{Z1}$ is the minimum according to Formula (2). At the same time, the mirror displacement in the direction of gravity and inclination angle as well as the mirror surface error reach the minimum. Neutral plane theory is used to estimate the optimal installation position of flexible structures.

Notably, the mirror surface error caused by the temperature fluctuation and non-flatness error are also affected by the installation position of the flexible support. The analysis results showed that a large distance between the rotation center and the optimal installation position corresponds to a high mirror surface error. As shown in Figure 9, this rule represents an approximate parabolic function [9]. However, optimal installation planes caused by temperature fluctuation and non-flatness error do not necessarily coincide with the neutral plane. The final installation position $L$ after a comprehensive comparison is 1.5 mm.

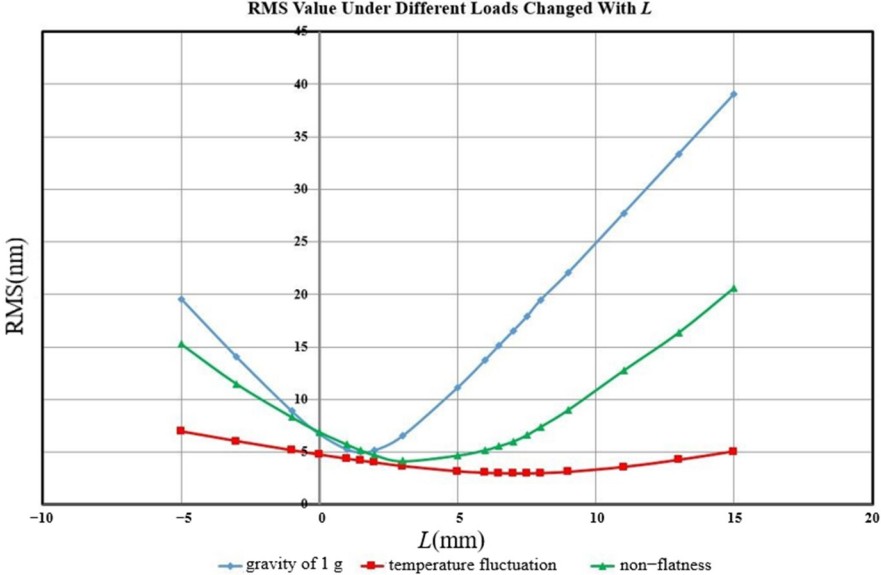

**Figure 9.** RMS value changed with distance $L$.

### 3.2. Static Analysis of Mirror Assembly

The parameterized model of the flexible support structure is established, and static analysis is conducted under different loads with finite element analysis software. Static analysis includes influences of gravity, temperature fluctuation, and non-flatness error

applied to the assembly. The performance of the flexible structure is evaluated by calculating the mirror surface peak-to-valley value (PV), root mean square (RMS) value, rigid displacement, and tilt angle. The surface error, rigid displacement, and inclination angle under gravity must be strictly controlled, especially when the assembly is placed vertically or horizontally. Furthermore, parts will deform differently when the temperature fluctuates because of different material thermal expansion coefficients and impacts on the mirror surface. Temperature fluctuation is usually set to 4 °C. The non-flatness error transfers to the mirror surface through flexible structures. Therefore, according to design requirements, the non-flatness error is usually set to 0.01 mm on one of the flexible support installation surfaces. Finally, the mirror assembly is subjected to the composite load of all above.

The simulation analysis results showed that the PV value under the composite load is 49.68 nm, the RMS value is 5.6 nm, displacement of the mirror is 0.005 mm, and inclination angle is 2″. The PV value under the garity is 51.6 nm, the RMS value is 5.01 nm, displacement of the mirror is 0.0037 mm, and inclination angle is 1.5″. The PV value under a temperature fluctuation of 4 °C is 22.1 nm, with an RMS value of 4.16 nm. The PV value under a non-flatness error of 0.01 mm is 21.65 nm, with an RMS value of 5.17 nm. The simulation analysis results are listed in Table 2.

**Table 2.** Simulation analysis results.

| Conditions | PV Value (nm) Results | RMS Value (nm) Results | Displacement (mm) Results | Inclination Angle (″) Results |
|---|---|---|---|---|
| Gravity of 1 g | 51.6 | 5.01 | 3.7 | 1.5 |
| Temperature fluctuation | 22.1 | 4.16 | —— | —— |
| Non-flatness error | 21.65 | 5.17 | —— | —— |
| Composite load | 49.68 | 5.6 | 5 | 2 |

Figure 10 shows the mirror surface displacement cloud picture under composite load. The mirror surface error cloud picture is presented in Figure 11. A coordinate system with the center of the mirror surface as the origin is established. The ratio of the x coordinate value of any point to the maximum diameter of mirror surface is defined as x/Rmax and the same for y/Rmax. The changed z coordinate value of any point on the mirror surface is defined as color distribution of cloud pictures.

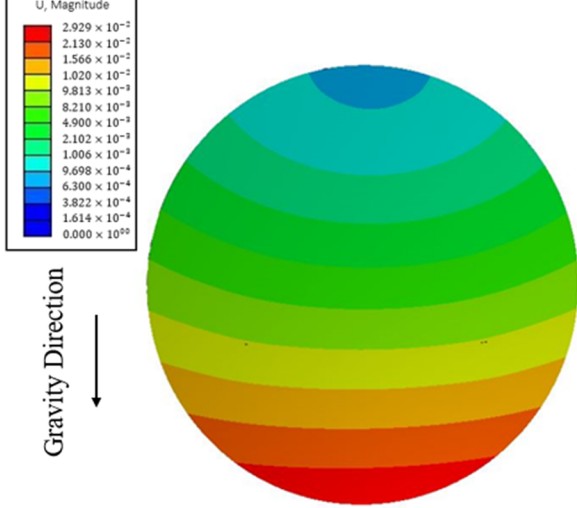

**Figure 10.** Mirror surface displacement cloud picture.

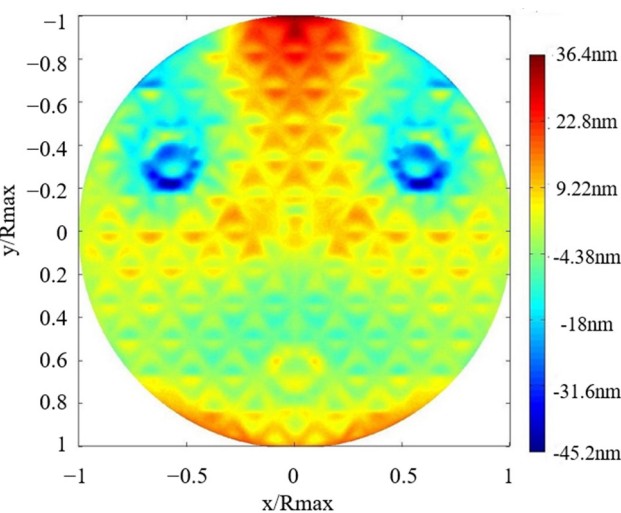

**Figure 11.** Mirror surface error cloud picture.

## 4. Dynamic Analysis of Three Flexible Supports

The dynamic investigation of the mirror assembly is generally divided into modal and strength analyses. Former fundamental frequencies of the mirror assembly, especially the first-order natural vibration mode, are considered in modal analysis to avoid damage caused by resonance. Strength analysis is used to evaluate the ability of the flexible support structure to bear complex vibration and find the weakest position of flexible supports [10].

### 4.1. Modal Analysis

Modal analysis can calculate former natural frequencies of the mirror assembly and is also the prerequisite for subsequent strength and fatigue analyses. The first three order modes of fundamental frequencies are of primary importance. The first 20 order modes of the mirror assembly are calculated using a finite element software to cover the frequency bandwidth of the vibration signal during experiment, transportation, and launching. The analysis results showed that the first three order frequencies of mirror components are 117.60, 117.71, and 119.75 Hz, which meet the design requirements. The first-order natural vibration mode shape rotates around the *x*-axis, which is a virtual coordinate axis perpendicular to the optical axis. The second-order natural vibration mode shape rotates around the *y*-axis, which is a virtual coordinate axis parallel to the optical axis. The third-order natural vibration mode shape rotates around the *z*-axis, which is a virtual coordinate axis determined by the right-hand rule. The vibration mode cloud picture of the first three orders is shown in Figure 12.

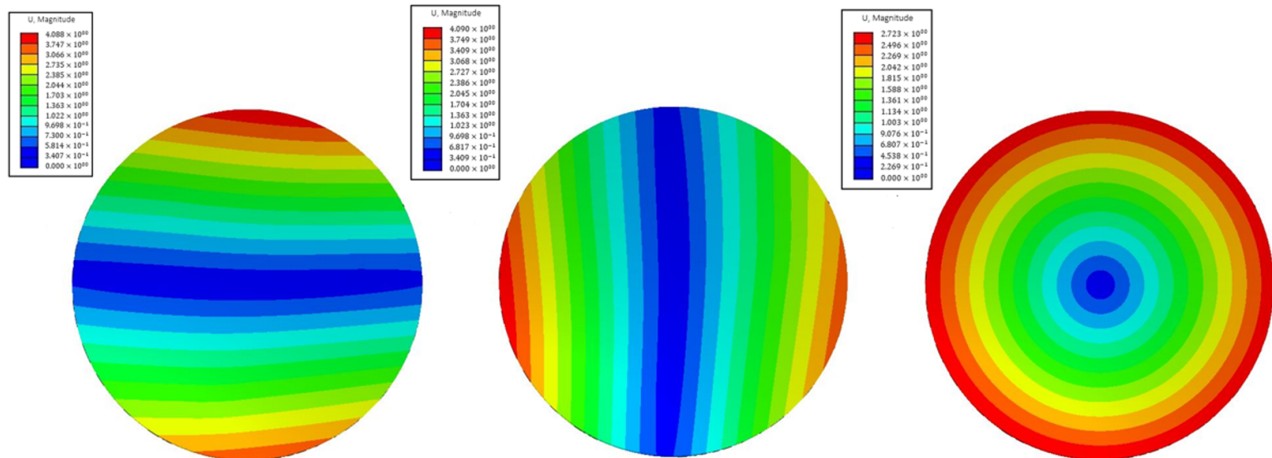

**Figure 12.** First three orders of the natural vibration mode cloud picture.

### 4.2. Strength Analysis

The input acceleration signal in strength analysis is usually in the form of sine or random vibration. Three stress and acceleration sampling points are set on the mirror surface, as shown in Figure 13. The coordinate system is the same as that in the finite element model. The corresponding magnification is calculated by comparing input and output acceleration values. The stress distribution is determined to evaluate the ability of the support structure to bear vibration. The test conditions of sine and random vibrations are listed in Tables 3 and 4, respectively. Strength analysis, especially the input PSD random vibration signal and the stress response, is the prerequisite for subsequent fatigue analysis.

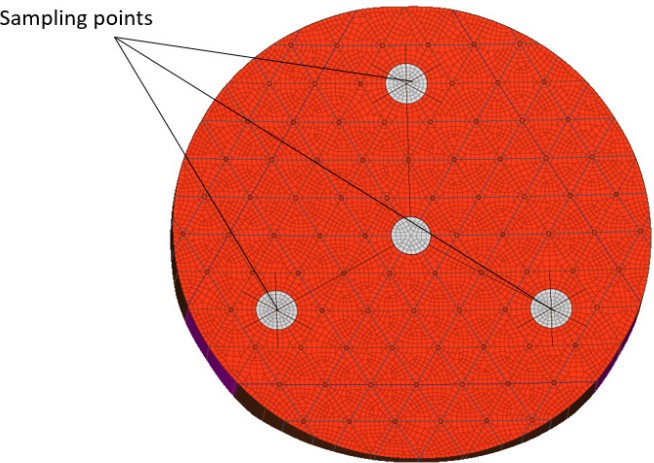

**Figure 13.** Location of sampling points.

**Table 3.** Sine vibration test conditions.

| Direction | *x*-axis | | *y*-axis | | *z*-axis | |
|---|---|---|---|---|---|---|
| | Frequency (Hz) | Magnitude | Frequency (Hz) | Magnitude | Frequency (Hz) | Magnitude |
| Parameters | 5–10 | 13.8 mm | 4–9 | 1.48 g | 4–9 | 1.48 g |
| | 10–15 | 5.5 g | 9–31 | 7.5 g | 9–31 | 7.5 g |
| | 15–31 | 7.5 g | 31–80 | 4 g | 31–80 | 4 g |
| | 31–80 | 5.5 g | 80–100 | 2.5 | 80–100 | 2.5 g |
| | 80–100 | 2.8 g | | | | |
| Scan frequency | 2 oct/min | | | | | |

**Table 4.** Random vibration test conditions.

| Direction | *x*-axis | | *y*-axis | | *z*-axis | |
|---|---|---|---|---|---|---|
| | Frequency (Hz) | Magnitude | Frequency (Hz) | Magnitude | Frequency (Hz) | Magnitude |
| Parameters | 10–20 | +6 dB/oct | 10–20 | +6 dB/oct | 10–20 | +6 dB/oct |
| | 20–125 | 0.075 g$^2$/Hz | 20–105 | 0.0315 g$^2$/Hz | 20–105 | 0.0315 g$^2$/Hz |
| | 125–185 | 0.01 g$^2$/Hz | 105–165 | 0.004 g$^2$/Hz | 105–165 | 0.004 g$^2$/Hz |
| | 185–200 | 0.075 g$^2$/Hz | 165–200 | 0.0315 g$^2$/Hz | 165–200 | 0.0315 g$^2$/Hz |
| | 200–2000 | −3 dB/oct | 200–2000 | −3 dB/oct | 200–2000 | −3 dB/oct |
| Total RMS acceleration | 6.69 g | | 4.33 g | | 4.33 g | |
| Testing time | 2 min | | | | | |

The analysis results showed that the maximum stress responses of sine vibration of 95, 114, and 126 MPa in the *x*-, *y*-, and *z*-axes, respectively, are at the position of clover grooves. The acceleration response amplification in the *x*-, *y*-, and *z*-axes is 23.93, 23.41, and 32 times, respectively. The maximum stress responses of random vibration of 185,

253, and 296 MPa are also at the position of clover grooves, and their respective total RMS acceleration response amplification is 1.34, 1.69, and 1.82 times, respectively.

## 5. Analysis of Fatigue Life

The random vibration load applied to the structure lasted for only a short time and the structure remains unaffected by fatigue when designing space optical remote sensors. However, the designing scheme becomes unacceptable when only traditional mechanical analysis is adopted. The stress and the temperature of mirror components will increase with vibration, while the fatigue limit of the flexible structure will decrease and cause fatigue failure.

### 5.1. Analysis of Vibration Fatigue

Many expression forms of the S–N curve are available for a certain material in the field of mechanics of materials. The S–N curve can be expressed in the form of a power function as $s^m \cdot N = C$, where $s$ is the critical stress and $m$ and $C$ are material constants that can be obtained through many experiments. Random vibration can be divided into two forms through the expression forms of input signals. If the peak probability density function of an input random vibration signal can be obtained via Formula (3), then we have the Rayleigh distribution. The amplitude probability density function $P(s_i)$ is similar to the peak probability density function. If the peak probability density function of an input random vibration signal can be obtained using Formula (4), then we have the Gaussian distribution. The amplitude probability density function $P(s_i)$ can be calculated according to Dirlik theory. Fatigue life can be derived from PSD for either case as follows:

$$P(s) = \frac{1}{\sigma^2} e^{-\frac{s^2}{2\sigma^2}}, \tag{3}$$

$$P(s) = \frac{1}{\sqrt{2\pi\sigma^2}} e^{-\frac{s^2}{2\sigma^2}}, \tag{4}$$

where $\sigma$ is the standard deviation.

In addition to PSD and the S–N curve, obtaining the stress and strain of the flexible structure extracted from the finite element analysis software or gathered from the random vibration experiment as well as surface roughness and residual stress when using fatigue analysis software is necessary. The fatigue analysis results showed that the random vibration fatigue life of the flexible structure is about 48,680,000 cycles. The structure is usually considered without damage when the fatigue life is more than 10,000,000 cycles. The dangerous position and the fatigue life cloud picture are presented in Figure 14.

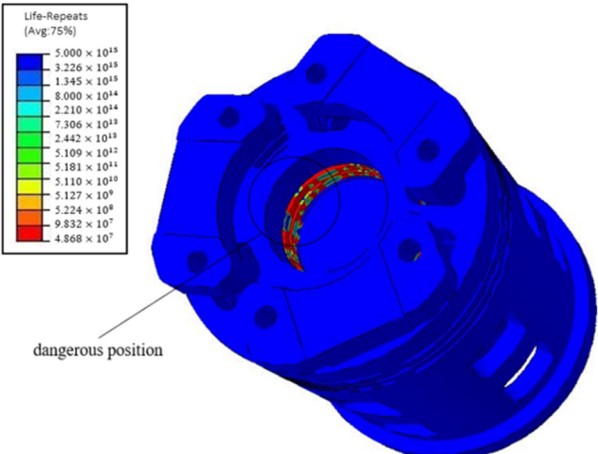

**Figure 14.** Fatigue life cloud picture.

### 5.2. Analysis of Thermal–Mechanical Coupling Fatigue

The heat of the mirror assembly increases with vibration loads. The S–N curve of the titanium alloy used in the flexible structure decreases significantly due to high temperature. Moreover, the fatigue limit of titanium alloy decreases due to thermal stress [11,12]. Fatigue failure will occur under the influence of thermal stress and random vibration load. Therefore, estimating the fatigue life of structures in the temperature field is necessary to ensure the reliability of the structure. The maximum temperature of the assembly during vibration is about 330 K. Amend the S–N curve of the titanium alloy according to the temperature and calculate the fatigue life of flexible structures. The results show that fatigue life of the flexible structure meets the design index at 21,790,000 cycles. Compared with the fatigue life of random vibration only, the thermal–mechanical coupling fatigue decreases sharply.

### 5.3. Analysis of Abrasion Fatigue

As shown in Figure 15, abrasion occurs on contact surfaces between the flexible structure and the rigid connector. Abrasion will loosen fasteners and cause fatigue damage [13]. Therefore, analyzing the abrasion of the contact surface is necessary. The non-flatness error is applied to one of the contact surfaces in front of the flexible structure to simulate abrasion. The surface roughness of the abrasion surface is considered beyond Ra6.3 in the subsequent fatigue analysis. The analysis results showed that the fatigue life of the flexible structure is about 45,000,000 cycles.

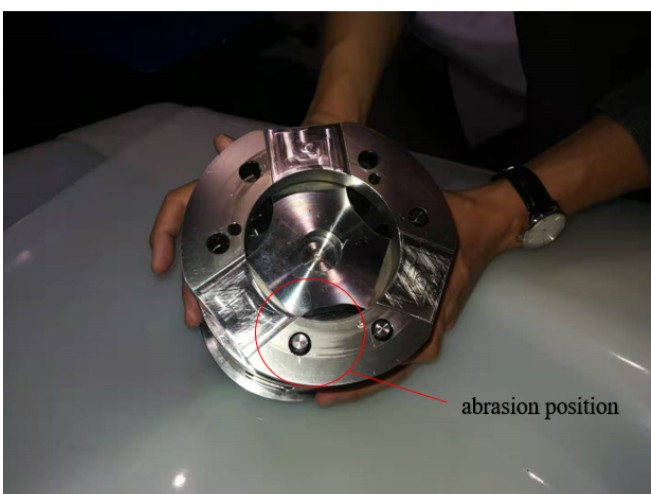

**Figure 15.** Abrasion of the flexible structure.

## 6. Environmental Tests

The real mirror assembly is different from the ideal model because of manufacturing and assembly errors. Therefore, environmental tests on the mirror assembly should be performed to verify the accuracy of the simulation results and eliminate the assembly internal stress. Environmental tests typically include vibration and temperature experiments. The mirror surface error, rigid displacement, and tilt angle of the mirror body directly reflect the performance of the mirror assembly. The mirror surface error must be detected with an interferometer during the process of experimenting and assembling, as shown in Figure 16. The mirror is placed horizontally and the optical axis is vertical during processing. The mirror surface is first polished without rigid connectors and flexible supports. Rigid connectors and flexible supports are installed when the mirror surface error RMS value is polished to nearly $\lambda/5$. The mirror is placed vertically and the optical axis is horizontal when the mirror is inspected. The PV value, the RMS value, and the interference cloud picture are recorded before and after the environmental tests.

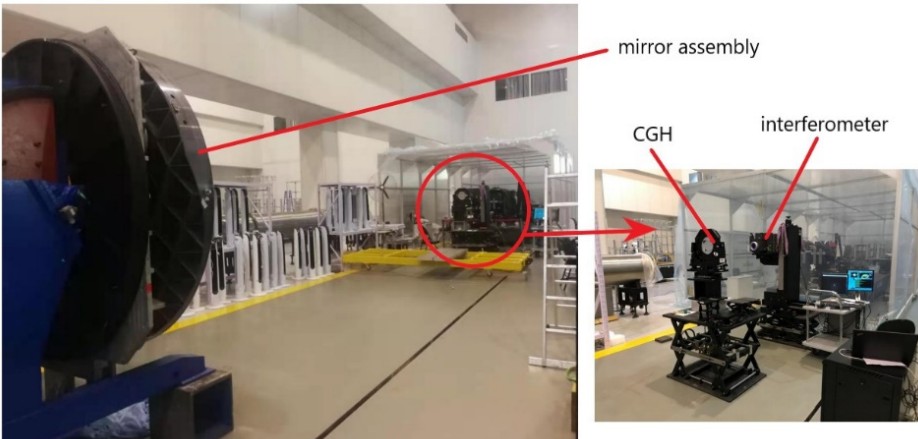

**Figure 16.** Detected mirror surface error using an interferometer.

The rigid displacement and the tilt angle of the mirror body are detected with a theodolite, as shown in Figure 17. Two flat mirrors are affixed on the edge of the mirror and on the test tooling. In addition, two flat mirrors should be in the same view to reduce system error. The tilt angle of the two flat mirrors is recorded and the rigid displacement of the mirror body is calculated before and after the environmental tests.

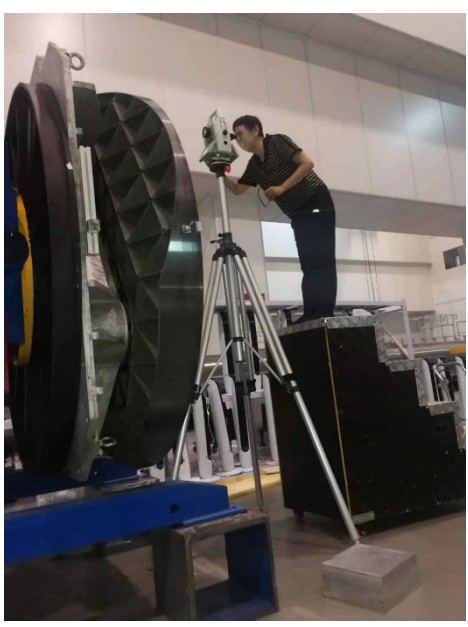

**Figure 17.** Detected rigid displacement and tilt angle of the mirror body using a theodolite.

### 6.1. Vibration Test

The vibration test is performed on a vibration experimental platform, as shown in Figure 18. The mirror assembly is connected to the experimental platform by a rigid tooling. Test input conditions of the vibration test are presented in Tables 3 and 4. Sine and random vibration tests are carried out in the $x$, $y$, and $z$ axes. The coordinate system is established on the basis of the vibration experimental platform. A 0.1 g sweep vibration test should be conducted before and after the sine vibration or random vibration test to eliminate the vibration stress and ensure that components are undamaged. Gauges (350 $\Omega$) are attached on the spring-leaf and clover flexible structures of flexible supports to collect the stress information during vibration.

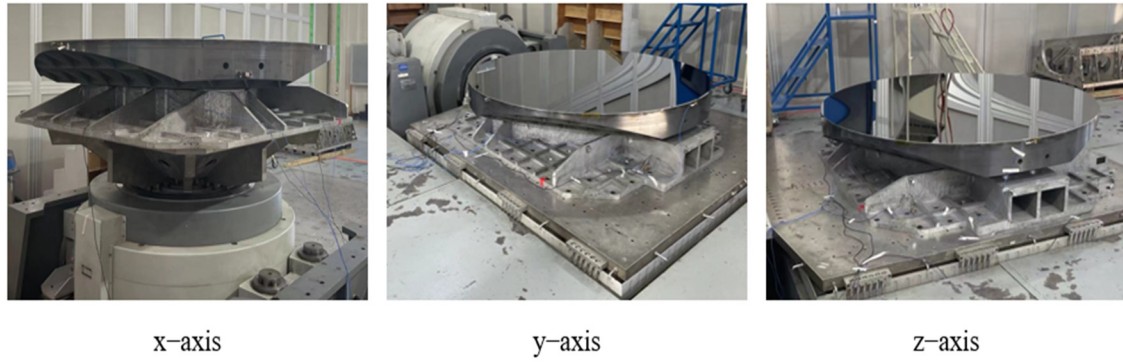

x−axis                   y−axis                   z−axis

**Figure 18.** Vibration test.

According to the vibration test results, the assembly fundamental frequency is 119.16 Hz. Acceleration responses of sine vibration are 23.57, 23.92, and 30.76 times in the *x*-, *y*-, and *z*-axes, and the maximum stress responses are 102.1, 107.4, and 119.7 MPa, respectively. The stress response curve of sine vibration at sampling points are presented in Figure 19. The total RMS acceleration responses of random vibration are 1.44, 1.62, and 1.86 times in the *x*-, *y*-, and *z*-axes, and the maximum stress responses are 176, 241, and 279 MPa, respectively. The stress response curves of random vibration at sampling points are illustrated in Figure 20. The test and simulation results are consistent, as shown in Tables 5–7.

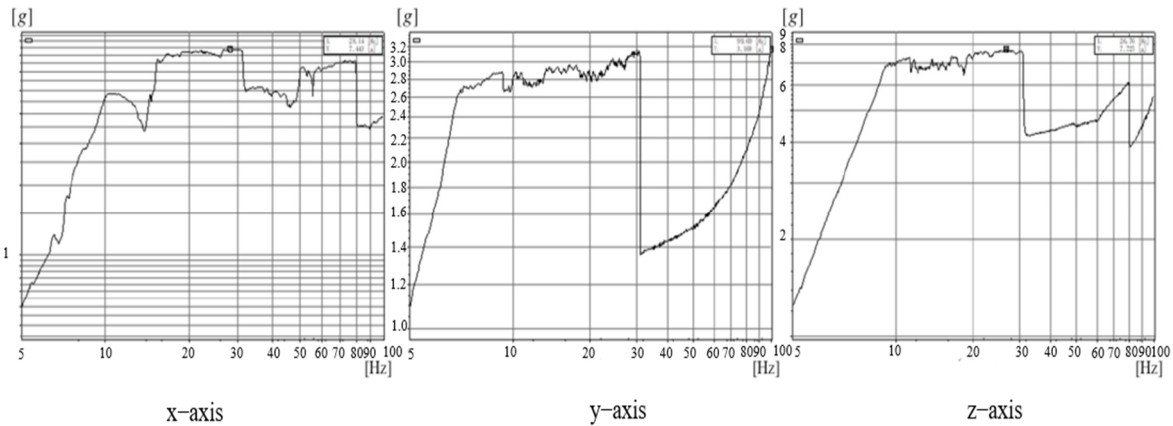

x−axis                   y−axis                   z−axis

**Figure 19.** Stress response of sine vibration at sampling points.

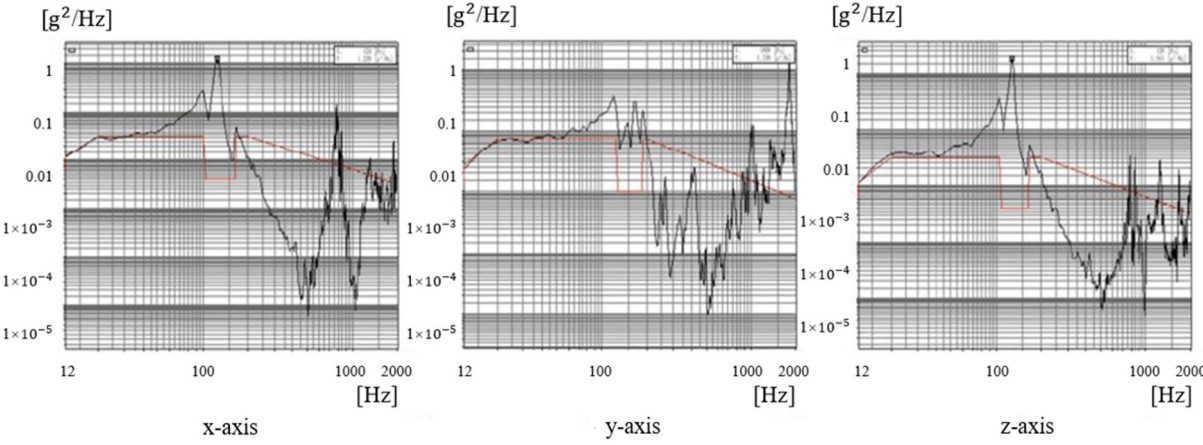

x-axis                   y-axis                   z-axis

**Figure 20.** Stress response of random vibration at sampling points.

**Table 5.** Comparison between test and simulation of acceleration amplification.

| Direction | Acceleration Response Amplification (Test) | Acceleration Response Amplification (Simulation) | Relative Error |
|---|---|---|---|
| *x* | 23.57 | 23.93 | 1.53% |
| *y* | 23.92 | 23.41 | 2.13% |
| *z* | 30.76 | 32 | 4.03% |

**Table 6.** Comparison between test and simulation of RMS acceleration amplification.

| Direction | RMS Acceleration Amplification (Test) | RMS Acceleration Amplification (Simulation) | Relative Error |
|---|---|---|---|
| *x* | 1.44 | 1.34 | 6.94% |
| *y* | 1.62 | 1.69 | 4.32% |
| *z* | 1.86 | 1.82 | 2.15% |

**Table 7.** Comparison between test and simulation of stress response.

| Project | Direction | Maxmum Stress Response (Test) | Maxmum Stress Response (Simulation) | Relative Error |
|---|---|---|---|---|
| Sine vibration | *x* | 102.1 MPa | 95 MPa | 6.95% |
| | *y* | 107.4 MPa | 114 MPa | 6.15% |
| | *z* | 119.7 MPa | 126 MPa | 5.26% |
| Random vibration | *x* | 176 MPa | 185 MPa | 5.11% |
| | *y* | 241 MPa | 253 MPa | 4.98% |
| | *z* | 279 MPa | 296 MPa | 6.09% |

Multiple measurements showed that the tilt angle of the mirror body changed by 3.4″, and the rigid displacement of the mirror body is 2.2 μm. Both values meet the design requirements. The mirror surface error RMS values before and after the vibration test are 0.019λ and 0.020λ, as shown in Figures 21 and 22, respectively. The test results showed that the vibration adaptability of the mirror assembly is satisfactory.

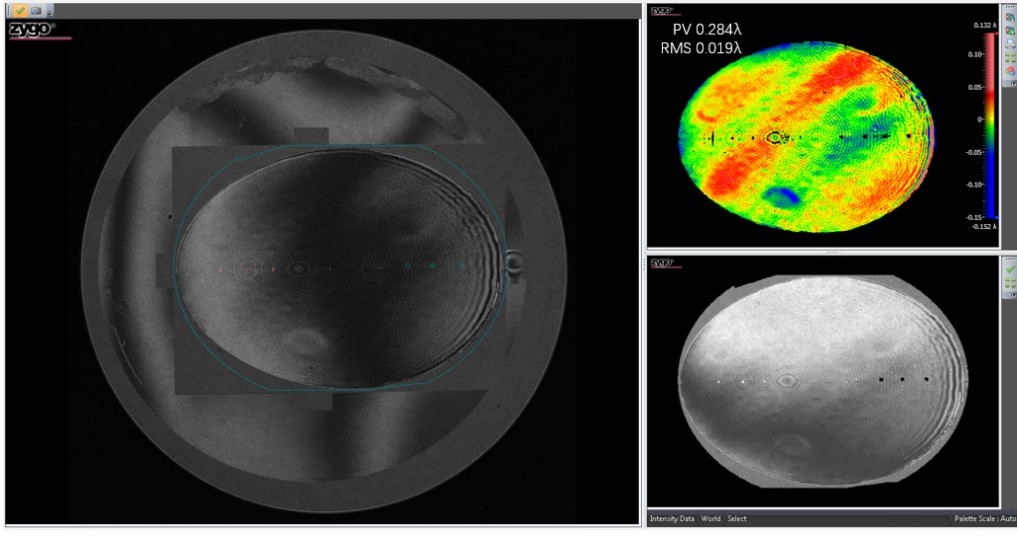

**Figure 21.** Mirror surface error before the vibration test.

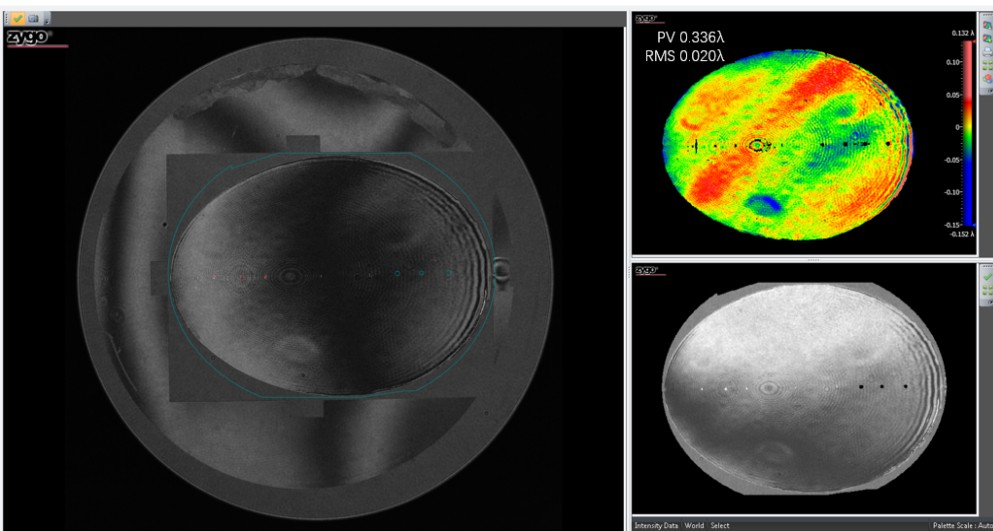

**Figure 22.** Mirror surface error after the vibration test.

### 6.2. Temperature Test

The temperature test reflects the mirror assembly under temperature fluctuation. The temperature of parts will increase without the help of temperature control components when the mirror is coated and subjected to the vibration test. Therefore, conducting a temperature test in a temp-enclosure as shown in Figure 23, in which a temperature load with a certain gradient is applied to the mirror assembly, is necessary. The temperature in the test begins at room temperature (20 °C), uniformly increases by 10 °C every 30 min, and is maintained for 2 h. The temperature drops uniformly by 10 °C every 30 min after it increases to 70 °C and is then maintained for 2 h until the temperature is equal to −30 °C. The temperature is subsequently raised from −30 °C to room temperature. This process is looped twice.

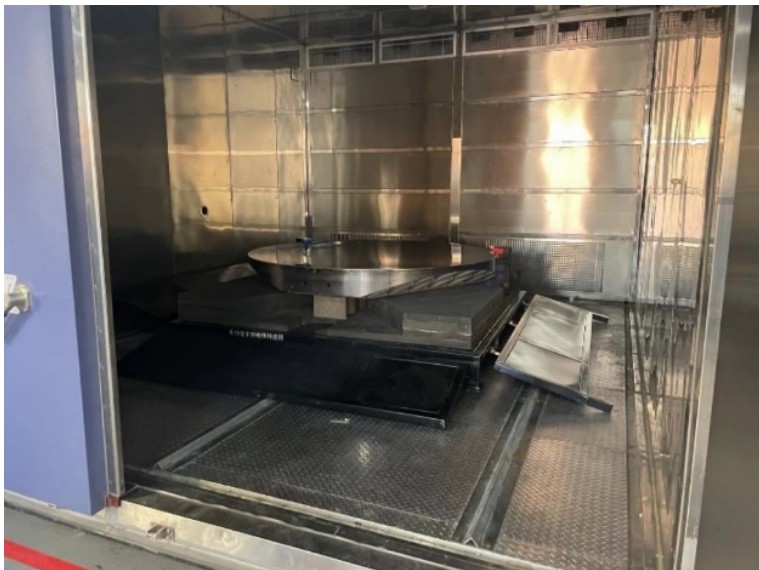

**Figure 23.** Temperature test.

Multiple measurements showed that the tilt angle of the mirror body changes by 1.5″, and the rigid displacement of the mirror body is 2.8 μm. Both values meet the design requirements. The mirror surface RMS values before and after temperature test are 0.020λ and 0.019λ, as shown in Figures 24 and 25, respectively. The test results showed the enhanced temperature adaptability of the mirror assembly.

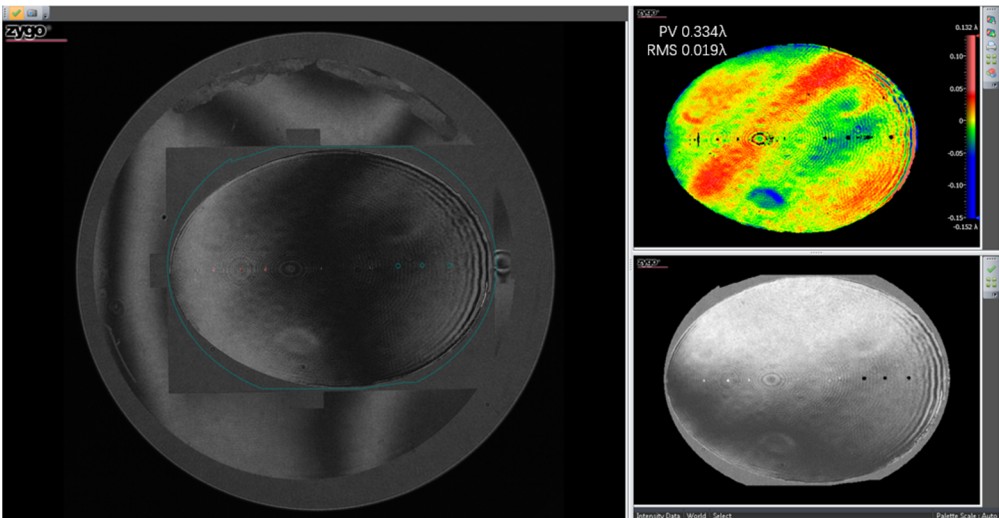

**Figure 24.** The mirror surface error before temperature test.

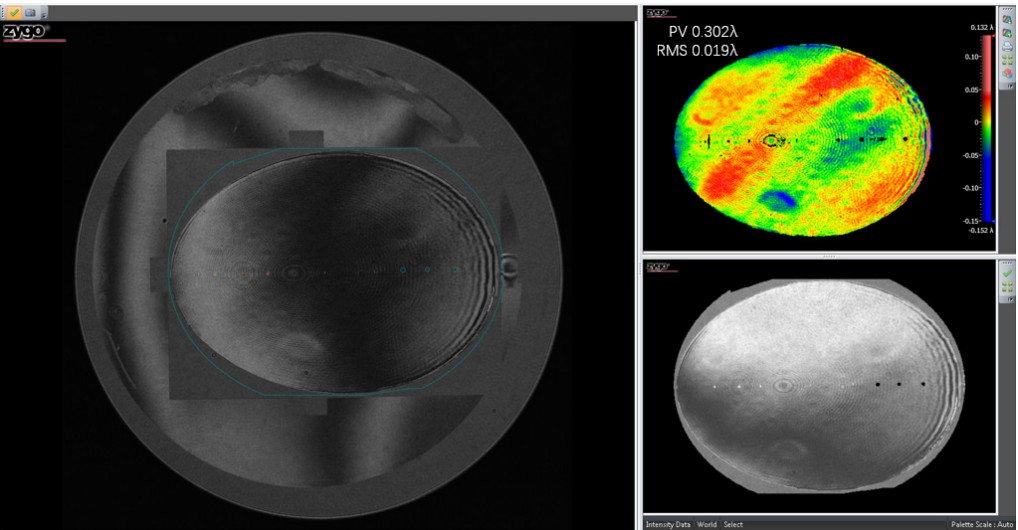

**Figure 25.** The mirror surface error after temperature test.

### 7. Conclusions

The final flexible support is designed and manufactured using the analysis mode and a series of tests. The mirror surface PV value is 49.68 nm and the RMS value is 5.6 nm under composited force. The flexible support fatigue life, which synthesized the thermal–mechanical, is more than 21,790,000 cycles. These features meet the design requirements. Experimental verification of the mirror assembly demonstrated that the first-order natural frequency of the mirror assembly is 119.16 HZ, which is consistent with the simulation result. The acceleration magnifications of sine vibration in the three axes are 23.57, 23.92, and 30.76 times. Stress responses of sine vibration at a dangerous point are 102.1, 107.4, and 119.7 MPa. The total RMS acceleration magnifications of random vibration in the three axes are 1.44, 1.62, and 1.86 times. Stress responses of random vibration at a dangerous point are 176, 241, and 279 MPa. These findings meet the design requirements and are consistent with the simulation results. The final flexible support design scheme is reliable. The analysis mode is proven to be efficient and accurate. The results of this study can provide a theoretical reference for designing other flexible supports.

**Author Contributions:** Project administration, K.W.; methodology, W.L. and J.X.; experiment and analysis, J.X.; writing—original draft, J.X.; and writing—review and editing, Q.H. and G.L. All authors have read and agreed to the published version of the manuscript.

**Funding:** This research is funded by the National Natural Science Foundation of China (NSFC) (11703027).

**Institutional Review Board Statement:** Not applicable.

**Informed Consent Statement:** Not applicable.

**Data Availability Statement:** Not applicable.

**Conflicts of Interest:** The authors declare no conflict of interest.

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
