# Peer review of "Design of a 2 m Primary Mirror Assembly Considering Fatigue Characteristics"

_applsci, doi:10.3390/app122010326_

Round 1
Reviewer 1 Report (New Reviewer)
This manuscript concerns to the design requirements of a 2 m mirror assembly installed on a large space optical remote sensor. The mirror body and rigid connectors are designed using topology and size optimization methods. The initial design scheme of flexible supports is then proposed through an in-depth exploration of fatigue failure mechanism caused by impacts of size parameters, thermal–mechanical vibration, and surface abrasion. The comprehensive analysis mode of the flexible support structure is established with the help of finite element and fatigue analysis software programs. The paper can be improved by considering the following issues:
1. Page 2, Section 2.1: Add details of the finite elements used (type, number of nodes, degrees of freedom).
2. Page 8, Figure 10: Add unit.
3. Page 9, Figure 12: Add unit.
4. Page 10, Line 223, Sentence: "The total RMS acceleration response curve of random vibration at sampling point #1 in the X-axis is illustrated in Figure 14." It would be easier for readers to read your work by providing the descriptions of sampling point on Figure 13.
5. Page 10, Figure 14: It's poor quality.
6. Page 14, Figure 20 and 21: Please correct the formatting the description.
7. Page 15, Conclusions: the conclusions in bulleted form are clearer to me.
Author Response
Dear reviewer:
Thanks for your suggestions on my work.
I have deleted some exaggerated self-citations and changed parts of some text. Also, I have changed some poor quality pictures and added some tables, pictures and details of the finite elements for a clearer conclusions.
Kind regards,
Jiakun Xu
Reviewer 2 Report (New Reviewer)
Other real life applications of very large size mirrors can be discussed. Also, cost of these mirrors and their replacement costs can be discussed.
Author Response
Dear reviewer:
Thanks for your suggestions on my work.
I have deleted some exaggerated self-citations and changed parts of some text. Also, I have changed some poor quality pictures and added some tables, pictures and details of the finite elements for a clearer conclusions.
Kind regards,
Jiakun Xu
Reviewer 3 Report (New Reviewer)
The present manuscript summarizes the results of modelling and testing on real the design requirements of a 2 m mirror assembly installed on a large space optical remote sensor. The design scheme of the mirror’s flexible supports was performed by exploring the fatigue failure mechanism caused by the impacts of size parameters, thermal–mechanical vibration, and surface abrasion. By using finite element and fatigue analysis software programs the reliability of the mirror assembly and the design of flexible supports was optimized. The article has potential to provide theoretical reference for designing other flexible supports. Even the subject of the article is aiming an extremely narrow / specialized field, I consider that the authors have exaggerated with the self-citations (approximately 35% of the total of 18 titles
Author Response
Dear reviewer:
Thanks for your suggestions on my work.
I have deleted some exaggerated self-citations and changed parts of some text. Also, I have changed some poor quality pictures and added some tables, pictures and details of the finite elements for a clearer conclusions.
Kind regards,
Jiakun Xu
This manuscript is a resubmission of an earlier submission. The following is a list of the peer review reports and author responses from that submission.
Round 1
Reviewer 1 Report
The problem could be stated in a form that better explains the usage of simulation tools and experiments. Figures 3 and 4 should avoid using colours, while figures from simulations should be of better quality with readable legends. Many figures are low quality and should improve. Major improvement should be made to conclusions, where the benefit of performed simulations should be highlighted. There are several minor errors (units) that should be corrected.
Reviewer 2 Report
see attached file
